# OpenReview forum: "Risk-adaptive Activation Steering for Safe Multimodal Large Language Models"
_ICLR.cc/2026/Conference — ICLR 2026 Conference Withdrawn Submission_

### Official Review · Reviewer_dkiV · 2025-10-30

**Soundness:** 2
**Presentation:** 3
**Contribution:** 2
**Rating:** 4
**Confidence:** 4

**Summary:**

This paper introduces Risk-adaptive Activation Steering (RAS), an inference-time method to improve multimodal LLM safety by reformulating queries, estimating risk via early-token similarity, and then adaptively steering activations toward refusal responses. The method aims to avoid high compute cost and over-refusal problems in prior safety-alignment approaches.

**Strengths:**

- Targets an important problem: scalable safety for MLLMs without retraining.
- Combines visual prompting, distribution-based risk scoring, and activation steering into a structured pipeline.
- Empirically effective: reduces attack success rate while preserving utility and improving inference speed.
- Ablation analysis supports core design choices (e.g., γ, early token count).
- Implementation is architecture-agnostic and training-free.

**Weaknesses:**

- Novelty is modest; core mechanism mainly integrates known techniques (vision prompts + logit similarity + activation steering).
- Significant reliance on prompting heuristics raises concerns about stability and generalization.
- Risk estimation based on early-token distributions lacks theoretical grounding and may be brittle in long-chain reasoning.
- Query reformulation may introduce failure cases on fine-grained visual tasks or adversarially perturbed images.
- Experiments focus on specific open-source models; unclear generalization to stronger proprietary models.

**Questions:**

1. How sensitive is RAS to prompt phrasing and visual context formatting?
2. Does early-token risk scoring degrade for long multi-turn or chain-of-thought interactions?
3. Can the authors provide concrete failure cases where benign queries receive high risk scores?
4. How does RAS interact with RL-aligned models (e.g., GRPO-trained MLLMs)?
5. Would multi-layer activation steering improve robustness vs. last-layer steering only?
6. Does the method handle image-based adversarial triggers or generated visual attacks?
7. Can the authors report detailed latency breakdown per stage, not only tokens-per-second?
8. How does query reformulation behave on fine-grained grounding tasks where added context may confuse perception?

---

### Official Review · Reviewer_x5Ew · 2025-10-30

**Soundness:** 3
**Presentation:** 3
**Contribution:** 3
**Rating:** 4
**Confidence:** 3

**Summary:**

This paper addresses a critical limitation of Multimodal Large Language Models (MLLMs): their vulnerability to malicious intent embedded in images (e.g., hidden harmful instructions) despite textual safety alignment. It identifies two core challenges with existing methods: training-based approaches are costly in data curation and computation, while inference-time methods either over-refuse benign queries (via safety prompts) or slow inference (via iterative response refinement). To solve these, the authors propose Risk-adaptive Activation Steering (RAS), an inference-time defense that operates in three stages: (1) vision-aware query reformulation appends concise visual summaries and safety prompts to strengthen cross-modal attention to safety-critical image regions; (2) exponentially weighted risk evaluation computes continuous risk scores by measuring similarity between the reformulated query’s output distribution and "refusal prototypes" (derived from unsafe text queries); (3) risk-adaptive activation steering adjusts model activations proportionally to the risk score—minimizing interference with benign queries while steering unsafe ones toward refusals. Extensive experiments across benchmarks (e.g., MM-SafetyBench, SPA-VL for safety; Sci-QA, MM-Vet for utility) and models (LLaVA-1.5, Qwen-VL-Chat) show RAS reduces attack success rates (ASR) by up to 89.5%, preserves general task performance, and improves inference speed compared to baselines like FigStep and ETA.

**Strengths:**

- Targeted Diagnosis of MLLM Safety Gaps: The paper provides analysis of why MLLMs fail at multimodal safety, which is insufficient cross-modal attention to safety-critical image regions. Using attention maps and Fisher Discriminant Ratio metrics, it demonstrates that models ignore harmful visual content but attend to harmful text (e.g., "bomb" in text), laying a strong theoretical foundation for RAS.
- Efficiency and Precision in Inference-Time Alignment: RAS avoids the pitfalls of prior methods: it eliminates iterative response refinement and uses continuous risk scores (instead of binary "safe/unsafe" judgments) to adapt intervention strength. This design ensures minimal interference with benign queries (utility preservation) while effectively blocking malicious ones, which is a balance missing in baselines like CoCA (over-reliance on logit calibration) or ECSO (iterative self-evaluation).

**Weaknesses:**

- **Limited Analysis of Visual Context Generation**: The paper mentions using a model to generate "concise visual contexts" (e.g., "The image shows illegal drug equipment") but provides little detail on how this generator is trained, validated, or its potential biases. If the visual context generator misdescribes safety-critical content (e.g., fails to identify a hidden weapon), RAS’s risk evaluation would be compromised, which is not explored.
- **Lack of Adversarial Robustness Testing for RAS Itself**: The paper evaluates RAS against existing jailbreak benchmarks (e.g., FigStep’s typographic prompts) but does not test whether adversaries could bypass RAS. For example, an attacker might manipulate images to confuse the visual context generator or craft queries that lower risk scores artificially, there is no analysis of such adaptive attacks is provided.
- **Oversimplification of "Safety-Critical" Visual Regions**: The paper focuses on discrete, object-like safety risks (e.g., bombs, drugs) but neglects ambiguous or contextual harms (e.g., images of self-harm, subtle hate symbols). It is unclear whether RAS can detect such nuanced visual threats, as the current risk evaluation relies on similarity to "refusal prototypes" trained on explicit unsafe text.

**Questions:**

See weaknesses

---

### Official Review · Reviewer_PYRi · 2025-10-31

**Soundness:** 3
**Presentation:** 3
**Contribution:** 3
**Rating:** 6
**Confidence:** 3

**Summary:**

The paper presents Risk-adaptive Activation Steering (RAS), an innovative inference-time alignment method aimed at enhancing the safety of multimodal large language models by dynamically adjusting visual attention to safety-critical regions in images. RAS addresses the challenge of ensuring that models provide helpful responses to benign queries while effectively refusing harmful ones. Through extensive experiments across various benchmarks, the authors demonstrate that RAS significantly reduces attack success rates without compromising general task performance or inference speed.

**Strengths:**

1. The proposed RAS method presents a novel way to enhance safety in multimodal large language models.
2. The paper includes extensive experiments across multiple benchmarks, demonstrating the effectiveness of the proposed method.
3. RAS significantly reduces attack success rates while maintaining general task performance, showcasing a strong balance between safety and utility while remaining efficient during inference.
4. The focus on strengthening visual attention to safety-critical regions addresses a critical gap in existing multimodal models.

**Weaknesses:**

The paper’s main weaknesses lie in its reliance on synthetic unsafe data, heuristic and unvalidated risk calibration, limited evaluation on real-world and diverse multimodal harms, and limited interpretability - related analysis.

**Questions:**

1. The approach focuses on enhancing visual attention to harmful regions in images. However, there is a scenario where both the image and the question are safe, while the models will still give harmful responses [1]. In this situation, does the proposed method still remain effective? It would be beneficial to explore where attention should be emphasized in such cases. Could you provide an explanation or example regarding this specific scenario?
2. The evaluation of safety benchmarks primarily focuses on ASR. It would be better to include results on the helpfulness of the model’s responses to demonstrate that the model does not excessively refuse benign queries.
[1] Safe Inputs but Unsafe Output: Benchmarking Cross-modality Safety Alignment of Large Vision-Language Models

---

### Official Review · Reviewer_Cj85 · 2025-11-02

**Soundness:** 3
**Presentation:** 2
**Contribution:** 3
**Rating:** 4
**Confidence:** 4

**Summary:**

The authors introduce Risk-adaptive Activation Steering (RAS), an inference-time framework for multimodal LLMs. The framework is divided into three stages. It appends a fixed safety prompt and a brief visual description of the image to the user's query, focusing attention on safety-critical regions. Using the reformulated input, it compares the output distributions of the first N tokens to unsafe prototypes to get a similarity and maps it to a risk score. When generating, it steers the last-layer activations of those first tokens toward the unsafe prototype by an amount proportional to the risk. This single-pass, early-token intervention substantially reduces jailbreak success while preserving utility and avoiding the overhead of other methods.

**Strengths:**

The authors introduce a novel framework for combining a safety-prompt + visual-context risk pass with risk-adaptive, early-token steering in a multimodal model. By design, the method considers both safety and utility. The authors support the design choices with preliminary experiments. The paper offers a good trade-off between utility and throughput compared to state-of-the-art methods.

**Weaknesses:**

I’m trying to understand the choice of N = 3, which seems somewhat restrictive. Is the strong performance mainly because the steer acts on early refusal templates like “I’m sorry,” effectively skewing many outputs to start that way? If so, I worry this could bias the model toward refusal and leave borderline cases unanswered. Could you comment on this? It seems N has a minor effect in the ablations because gamma is already fixed and could be ablated per se. This could also explain the marginal gains over “binary” in the ablations.

The paper benchmarks mainly against prompting and judge-and-regenerate defenses, but provides limited head-to-head comparisons with closely related activation-steering methods (for example: Steering Away from Harm: An Adaptive Approach to Defending Vision
Language Model Against Jailbreaks).

**Questions:**

My understanding is that the visual description, along with the safety prompt, is used only for risk evaluation, while the final answer is generated from the original image and query with risk-adaptive steering. Is this correct? For completeness, it would be helpful to include stage-1 ablations that isolate the safety prompt and the visual context.

In Figure 6, many of the unsafe examples are not covered by the sigmoid. Could you provide an explanation of this?

Could you please provide an explanation of why ECSO has better relative throughput in Figure 7?

Please provide details about the weakness highlighted above.

**Details Of Ethics Concerns:**

The concerns were addressed in the Ethics Statement Section of the paper.

---

### Note · Authors · 2025-11-13

I have read and agree with the venue's withdrawal policy on behalf of myself and my co-authors.